

# PKMYT1 knockdown inhibits cholesterol biosynthesis and promotes the drug sensitivity of triple-negative breast cancer cells to atorvastatin

Wei Gao[1,*], Xin Guo[2,*], Linlin Sun[3], Jinwei Gai[3], Yinan Cao[4] and Shuqun Zhang[1]

[1] Department of Oncology, The Second Affiliated Hospital of Xi'an Jiaotong University, Xi'an, China
[2] Department of Breast Surgery, The First Affiliated Hospital of Dalian Medical University, Dalian, China
[3] Day Surgery Center, Dalian Municipal Central Hospital, Dalian, China
[4] Graduate School of Dalian Medical University, Dalian, China
* These authors contributed equally to this work.

Corresponding author
Shuqun Zhang,
shuqun_zhang1971@163.com

## ABSTRACT

Triple negative breast cancer (TNBC) as the most aggressive molecular subtype of breast cancer is characterized by high cancer cell proliferation and poor patient prognosis. Abnormal lipid metabolism contributes to the malignant process of cancers. Study observed significantly enhanced cholesterol biosynthesis in TNBC. However, the mechanisms underlying the abnormal increase of cholesterol biosynthesis in TNBC are still unclear. Hence, we identified a member of the serine/threonine protein kinase family PKMYT1 as a key driver of cholesterol synthesis in TNBC cells. Aberrantly high-expressed PKMYT1 in TNBC was indicative of unfavorable prognostic outcomes. In addition, PKMYT1 promoted sterol regulatory element-binding protein 2 (SREBP2)-mediated expression of enzymes related to cholesterol biosynthesis through activating the TNF/ TNF receptor-associated factor 1 (TRAF1)/AKT pathway. Notably, downregulation of PKMYT1 significantly inhibited the feedback upregulation of statin-mediated cholesterol biosynthesis, whereas knockdown of PKMYT1 promoted the drug sensitivity of atorvastatin in TNBC cells. Overall, our study revealed a novel function of PKMYT1 in TNBC cholesterol biosynthesis, providing a new target for targeting tumor metabolic reprogramming in the cancer.

## INTRODUCTION

Breast cancer (BC) is a heterogeneous disease that can be divided into three primary categories of tumors, including hormone receptor (HR)-positive, HER2-enriched, and triple negative according to the levels of estrogen and progesterone receptors and HER2 amplification or overexpression (*Vagia, Mahalingam & Cristofanilli, 2020*). Notably, triple negative breast cancer (TNBC) is characterized by the absence of hormone receptor

expression (ER/PR) and the presence of a high level of human epidermal growth factor (HER2). Epidemiological evidence indicates that TNBC predominantly presents in women under the age of 40 before menopause, accounting for about 15–20% of all breast cancer cases (*Morris et al., 2007*; *Jiang et al., 2023*). Currently, chemotherapy is the major systemic therapy, but the efficacy of conventional postoperative adjuvant radiotherapy is poor (*Zhang et al., 2022*). In addition, some countries use combined used of drugs to treat TNBC. However, patient survival time has not been significantly improved (*Wang et al., 2024*). Accordingly, there is a need to actively develop new therapeutic regimens and targets for the treatment of TNBC.

Metabolic reprogramming is a hallmark for cancers. Lipids including steroids, isoprenoids, acylglycerols and phospholipids are the components of biological membranes that can be used for energy metabolism and storage and play important roles as signaling molecules (*Mayengbam et al., 2021*; *Koundouros & Poulogiannis, 2020*). Abnormal lipid metabolism is closely involved in the malignant processes of tumors. One of the characteristic changes is the abnormal increase of cholesterol biosynthesis. The accumulation of cholesteryl esters in tumor cells is associated with cell proliferation, metastasis, survival and drug resistance (*Wang et al., 2017*; *Guillaumond et al., 2015*). Furthermore, aberrant regulation of cholesterol biosynthetic flux or dysregulation of pathway enzyme expression is a driving factor of tumorigenesis. TNBC is categorized into three heterogeneous metabolic pathway-based subtypes (MPSs) with different metabolic profiles, including MPS1, MPS2, and MPS3. The MPS1 subtype is marked by a considerable increase in fatty acid, cholesterol, and steroid synthesis and a high rate of mutations in the PI3K and RTK-RAS pathways. In comparison to the MPS2 and MPS3 subtypes, MPS1 cell lines are more sensitive to lipid synthesis inhibitors (*Gong et al., 2021*). However, high toxicity of currently used inhibitors of lipid synthesis, such as C57 and fatostatin, prevents a wider practical application. Hence, to discover novel therapeutic targets for lipid and cholesterol biosynthesis has crucial significance.

Transcription factors such as SREBFs (especially SREBF2) and LXRs regulate cholesterol biosynthesis (*Goldstein & Brown, 2015*; *Nazih & Bard, 2020*; *Clendening et al., 2010*; *Sato, 2010*). SREBF2 is cleaved by proteases at low cholesterol levels in the endoplasmic reticulum, and the cleaved N-terminal portion subsequently translocates to the nucleus and binds to sterol response elements (SREs), activating the production of cholesterol biosynthesis enzymes such as HMGCR and SQLE (*Wang & Tontonoz, 2018*). Additionally, SREBF2 is a downstream factor and effector of numerous oncogenic signals, including the pRb, c-Myc, PI3K-AKT, and mTORC1 pathways (*Mullen et al., 2016*). Although small molecule inhibitors of SREBFs such as Fatostatin are effective in killing cancer cells, their toxicity significantly limits their practical application (*Li et al., 2014*). As a competitive inhibitor of HMGCR, statins are an enzyme that catalyzes HMG-CoA to mevalonate to inhibit cholesterol biosynthesis. Statins are the first-line therapy for hypercholesterolemia and play critical roles in preventing cardiovascular diseases. Statins could suppress the proliferation, invasion, and metastasis of various cancers, including breast cancer, and increase apoptosis in these tumors (*Wong et al., 2002*; *Jakobisiak & Golab, 2010*; *Sassano & Platanias, 2008*; *Fritz, 2009*). It has been found that statins inhibit the mevalonate/cholesterol synthesis pathway. In particular, statins could modulate

phosphoinositide 3-kinase/Akt serine/threonine kinase 1 and inflammatory signaling pathways and alter the expression of genes involved in lipid metabolism (*Gong et al., 2017*; *Dickerman et al., 2019*). However, the mechanisms underlying the anticancer effects of statins in TNBC remain poorly understood. WEE1 G2 checkpoint kinase (WEE1), membrane-associated tyrosine- and threonine-specific cdc2-inhibitory kinase (PKMYT1), and WEE2 oocyte meiosis inhibiting kinase (WEE1B) are the three conserved serine/ threonine kinases of the WEE1 kinase family. WEE1/PKMYT1 enzymes are involved in various biological functions and have varied functions in both normal and cancerous cells. PKMYT1 and WEE1 are essential regulators of the cell cycle, particularly during mitotic entry (*Asquith, Laitinen & East, 2020*; *Ghelli Luserna di Rorà et al., 2020*). Thus, they could act as tumor suppressor genes (*Ghelli Luserna di Rorà et al., 2020*). A previous study performed genome-wide CRISPR screening on a total of 563 cancer cell lines and found that WEE1 and PKMYT1 are required for cell viability in almost all the cell lines (*Liu et al., 2020*). A recent study identified PKMYT1 as the only member of the WEE family of kinases that is abnormally elevated in breast cancer tissues, in which it plays a critical role in the G2/M phase transition (*Schmidt et al., 2017*). However, the role of PKMYT1 as a serine/threonine kinase in tumor metabolic reprogramming remains unclear. In this study, PKMYT1 was identified as a critical driver of cholesterol biosynthesis in TNBC. PKMYT1 silencing prevented statin-induced SREBF2-dependent feedback activation, thereby significantly inhibiting tumor cholesterol biosynthesis and cell proliferation. Overall, our study identified PKMYT1 as a key regulator of TNBC cholesterol biosynthesis, indicating that it may be an effective therapeutic target.

## MATERIALS AND METHODS

### The UALCAN database

The UALCAN website provides a thorough analysis of cancer omics data and is accessible at http://ualcan.path.uab.edu. We utilized this platform to assess the mRNA expression of PKMYT1 in breast cancer. In addition, we evaluated the mutation frequency of PKMYT1 in different cancer types.

### Antibodies

Anti-PKMYT1(MA5-25029) was purchased from Invitrogen. Anti-Vinculin(sc-73614) was purchased from Santa Cruz Biotechnology (Dallax, TX, USA). Anti-HMGCR (ab174830) was purchased from Abcam (Abcam, Cambridge, UK). Anti-DHCR24 (10471-1-AP), Anti-SQLE (12544-1-AP), Anti-HMGCS1 (17643-1-AP), Anti-FDFT1 (13128-1-AP), Anti-TRAF1 (45D3) and Anti-SREBF2 (28212-1-AP) were commercially obtained from Proteintech (Proteintech Group, Inc., Rosemont, IL, USA). Anti-GAPDH (#5174S) and Anti-P-AKT (Ser473) (cas#4060S) were commercially obtained from Cell Signaling Technology (CST, Danvers, MA, USA).

### Cell culture

MDA-MB-453 cells were purchased from the Cell Bank of the Committee on Type Culture Collection of the Chinese Academy of Sciences (Shanghai, China). Dulbecco's modified

Eagle's medium (DMEM, GIBCO, Carlsbad, CA, USA) dissolved with 10% FBS (Sigma-Aldrich, St. Louis, MO, USA) was used to culture the cells in a humid incubator with 5% $CO_2$ at the temperature of 37 °C.

## Plasmids

PKMYT1 was subcloned into pBoBi expression vectors (Verma laboratory, Salk Institute La Jolla, CA, USA) and Flag-tagged expression vectors (Sigma-Aldrich, St. Louis, MO, USA). Plasmid DNA was mixed with serum-free medium and cells were transfected with Lipofectamine 2000 (Invitrogen) according to the manufacturer's instructions. Subsequently, the cells were incubated for 4 h at 37 °C with 5% $CO_2$. After the transfection, the cells were observed under a fluorescence microscopy (Zeiss, Oberkochen, Germany).

## Lentiviral shRNA cloning and stable cell line generation

The pLKO.1 within the AgeI/EcoRI sites at the 3′ end of the human U6 promoter was cloned with lentiviral shRNAs to target the genes of PKMYT1. The targeted sequences we as follows:

PKMYT1-sh1 5′-TGCGTTCTGTCCTTGTCATGA-3′, PKMYT1-sh2 5′-CTCCTACGGAGAGGTCTTCAA-3′, TRAF1-sh1 5′-GCCTTCTACACTGCCAAGTAT-3′. Subsequently, lentiviral particles containing these shRNAs were produced using a lentiviral packaging system. Specifically, the constructed pLKO.1-shRNA plasmid was co-transfected with packaging plasmids into HEK293T cells to collect and concentrate the viral particles for 48–72 h. The pLKO-BSD-shTRAF1 and pLKO-Puro-shPKMYT1 lentiviral particles were then used to transduce MDA-MB-453 cells, which were pre-treated with polybrene to enhance transduction efficiency. The cells were selected with antibiotic-containing medium 24 h after the transduction. The pLKO-BSD-shTRAF1-transduced cells and pLKO-Puro-shPKMYT1-transduced cells were selected with 6μg/ml Blasticidin and 2 μg/ml Puromycin, respectively. Finally, the knockdown efficiency of PKMYT1 and TRAF1 genes was verified using quantitative real-time PCR and Western blot to ensure the effectiveness of the shRNAs and the stability of shRNA-expressing cell lines for subsequent experiments.

## EdU assay and plate colony formation experiments

In EdU proliferation experiments, cells were first seeded in 24-well plates at the appropriate density and treated with complete medium containing 10 μM EdU for 2–4 h after attachment. Subsequently, the cells were fixed with 4% formaldehyde, permeabilized with 0.5% Triton X-100 and then passed through the Click-iT EdU Alexa Fluor 594 Imaging Kit (Invitrogen, Life Technologies, Carlsbad, CA, USA) according to the instructions. Ultimately, the proliferative capacity of the cells was assessed by calculating the proportion of EdU-positive cells to total cells under a fluorescence microscopy.

For plate colony formation experiment, the cells were placed in a 12-well dish (with 500 cells in each well) filled with DMEM containing 10% FBS from HyClone and 1% PS from HyClone and then incubated for 14 days. Next, the colonies were washed with PBS, treated with 4% paraformaldehyde for 30 min, and stained with Giemsa stain from Leagene for

another 30 min. The efficiency of the colonies was subsequently calculated by the formula: Colony formation efficiency = (number of clones/number of inoculated cells) × 100%.

## Western blot

Briefly, radioimmunoprecipitation assay (RIPA) buffer with phosphatase inhibitors (Roche, Sigma-Aldrich, St. Louis MO, USA) and protease inhibitors (Sigma-Aldrich, St. Louis, MO, USA) was performed to lyse the cells. After using sodium dodecyl sulfate–polyacrylamide gel electrophoresis (SDS-PAGE) for protein separation, the proteins were moved to PVDF membrane (Bio-Rad, Hercules, CA, USA). Next, specific primary antibodies and peroxidase-conjugated secondary antibodies were used to probe the proteins. Vinculin antibody was used for monitoring equal protein-sample loading. Visualization of the bands was realized using chemiluminescence.

## RNA-Seq

The SMART-RNAseq Library Prep Kit (AT4201, Hangzhou KaiTai) was employed to develop sequencing libraries based on total RNAs. Briefly, mRNA was isolated from total RNA using Sera-Mag Magnetic Oligo(dT) particles and then chemically fragmented. The fragmented RNA was reverse-transcribed into cDNA using random primers containing a tagging sequence at their 3′ ends. The cDNA libraries were amplified by the KAPA high-fidelity DNA polymerase. The quality of the libraries was assessed using the 2100 Bioanalyzer (Agilent Technologies, Santa Clara, CA, United States) to confirm that they met the required standards. High-throughput sequencing was performed on the NovaSeq 6000 platform (Illumina) with a sequencing depth of 50 million reads per sample using paired-end (PE) reads of 150 base pairs (bp) each. The sequencing libraries were not stranded.

To determine gene expression, the reads for each library were mapped using hisat2 (version 4.8.2) and then transformed to fragments per kilobase of exon per million fragments mapped (FPKM) using Cuffdiff 2.1.137. For differential gene expression analysis, the DESeq2 pipeline was employed. ClusterProfiler was used to perform biological pathway analysis. Professional technical staff at Hangzhou KaiTai Bio-lab assisted in performing RNA sequencing and library construction. The statistical power of this experimental designcalculated using RNASeqPower was 0.9996503 (sequencing depth: 50 million reads; number of biological replicates: 10; coefficient of variation: 0.5; effect size: 4; alpha: 0.01).

## Measure of cholesterol

After washing the cells with PBS, the cells were lysed with RIPA lysis buffer containing protease inhibitors and centrifuged at 4 °C to remove cell residue. Subsequently, the clear lysate was mixed with the enzyme reaction mixture using the Micro Total Cholestenone (TC) Content Assay Kit (Beijing Solarbio Science & Technology Co., Ltd, China) in accordance with the instructions, and the absorbance was read in a micro-enzyme labeler. Finally, the TC content in the sample was calculated from the standard curve.

### IC50 of atorvastatin

Drug sensitivity test was performed to directly assess the ability of cells to respond to drug-mediated pathway alterations. By comparing the sensitivity of PKMYT1 knockdown cells and control cells to atorvastatin, we were able to indirectly assess the effect of PKMYT1 on the cholesterol synthesis pathway. The IC50 value is a commonly used pharmacological metric to characterize the concentration of a compound (usually a drug) required to inhibit half of a biological or biochemical activity. In particular, the IC50 value is used to assess the magnitude of a drug's potency, with a lower value indicating a smaller concentration of the drug required to inhibit the target activity and a greater potency of the drug (*Park et al., 2022*). The IC50 of atorvastatin was detected with CCK-8 assay kit (Dojindo, Kumamoto, Japan). Specifically, $5 \times 10^4$ cells were seeded into 96-well plates and cultured for 24 h before adding various concentrations of atorvastatin (0.1 to 100 µM). After the addition of the drug, the cells were incubated for another 24 h. According to the instructions of the CCK-8 assay kit, 10µL of the CCK-8 solution was added to each well, and the plates were incubated at 37 °C for 2 h. Subsequently, the absorbance of each well was measured at a wavelength of 450 nm using a microplate reader. The cell viability at each concentration was calculated based on the absorbance values, and the IC50 value of atorvastatin was determined using nonlinear regression analysis in GraphPad Prism software.

### Statistical analysis

Before performing ANOVA, the data normality was assessed using Shapiro-Wilk test. ANOVA test was performed in order to compare the differences in means between three or more groups. Two-group differences were analyzed using unpaired and two-tailed Student's t-test. Data were expressed as mean ± standard error of the mean (SEM). ImageJ (version no.: 1.8.0_112; https://imagej.net/ij/) was used to normalize all of the relative protein expressions. The SPSS 18.0 software package (SPSS, Inc., Chicago, IL, USA) was used to perform statistical analyses.

## RESULTS

### PKMYT1 expression was upregulated in TNBC and promoted cell proliferation

To clarify the biological function of PKMYT1 in TNBC, we performed bioinformatics analysis using online databases and found that PKMYT1 had the highest alteration rate in invasive breast cancer and was primarily characterized by amplification (Fig. 1A). Subsequent analysis revealed that the *PKMYT1* expression was significantly higher in TNBC than in other breast cancer subtypes (Fig. 1B). Survival analysis revealed that progression-free survival of TNBC patients with high-expressed *PKMYT1* was relatively shorter (Fig. 1C). Subsequently, we developed TNBC cell lines with stable knockdown of PKMYT1 in MDA-MB-453 cells. It was found that the cell proliferation and clone formation were significantly suppressed by knocking down PKMYT1 (Figs. 1D and 1E). These data suggested that PKMYT1 played an oncogenic role in TNBC, but the exact mechanism remained to be investigated.

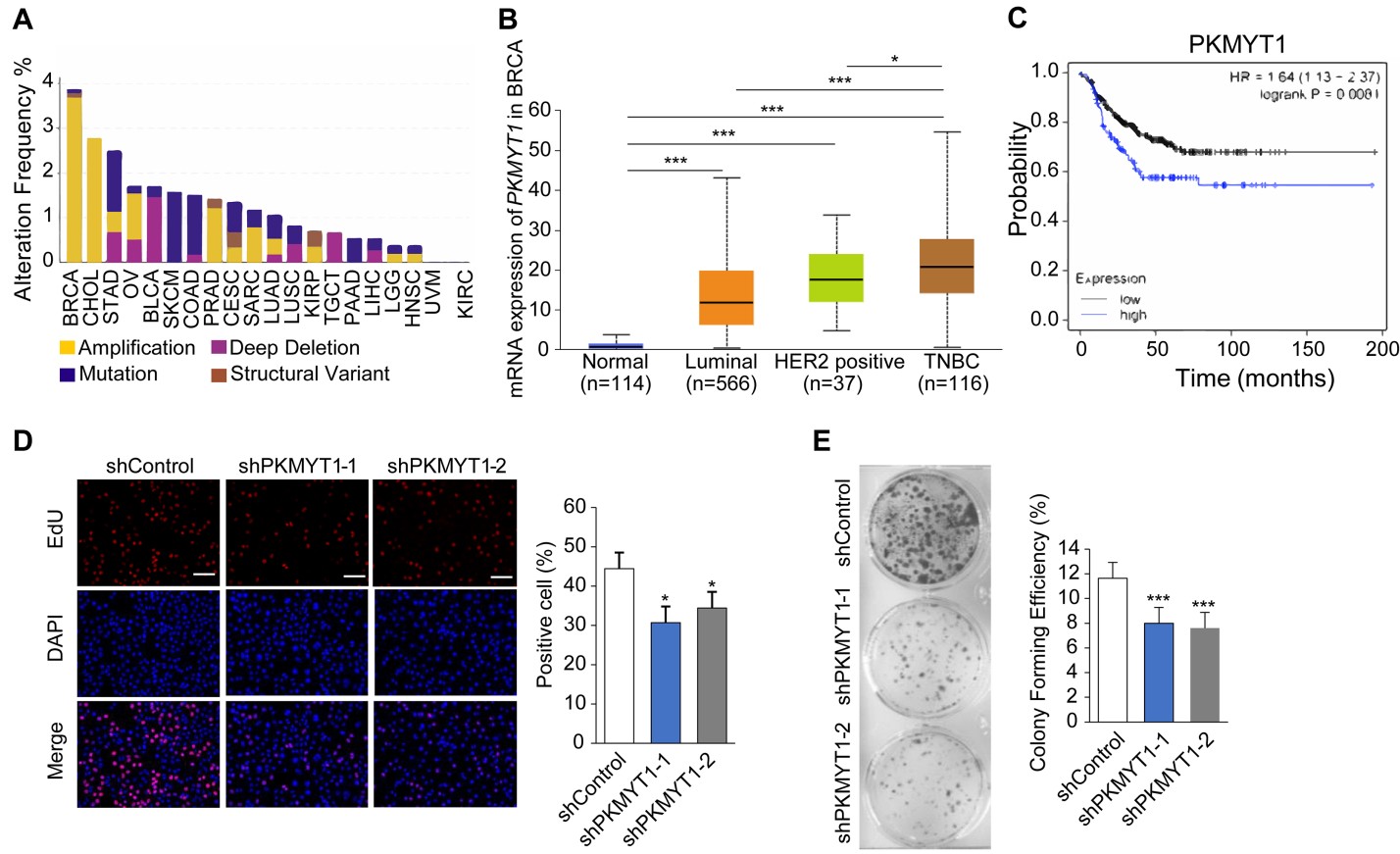

**Figure 1 PKMYT1 is highly expressed in TNBC and promotes cell proliferation.** (A) Mutation landscape of PKMYT1 based on TCGA database. (KIRC, kidney renal clear cell carcinoma; CHOL, cholangio carcinoma; KIRP, kidney renal papillary cell carcinoma; LUSC, lung squamous cell carcinoma; BRCA, breast invasive carcinoma; BLCA, bladder urothelial carcinoma; STAD, stomach adenocarcinoma; COAD, colon adenocarcinoma; SKCM, skin cutaneous melanoma; LUAD, lung adenocarcinoma; OV, ovarian serous cystadenocarcinoma; PAAD, pancreatic adenocarcinoma; PRAD, prostate adenocarcinoma; CESC, cervical squamous cell carcinoma and endocervical adenocarcinoma; LGG, brain lower grade glioma; LIHC, liver hepatocellular carcinoma; UVM, uveal melanoma; TGCT, testicular germ cell tumors; SARC, sarcoma; HNSC, head and neck squamous cell carcinoma) (http://www.cbioportal.org/index.do) (B) PKMYT1 mRNA expression levels in BRCA based on breast cancer subclasses. (C) Kaplan-Meier curves of the survival analysis (progression-free survival) of TNBC patients with PKMYT1 mRNA high and low expression. (red presents high expression, *n* = 122; black presents low expression, *n* = 270) (http://kmplot.com/analysis/index.php?p=service&start=1) (D) Lentiviral transfection was performed to generate MDA-MB-453 cells with stable knockdown of PKMYT1, and EdU staining assays were performed. *$^*P < 0.05$, ANOVA Test. (E) Clone formation assays were performed in MDA-MB-453-shPKMYT1-1/2 cells. $^{***}P < 0.001$, ANOVA Test.

## PKMYT1 promoted cholesterol biosynthesis in TNBC

We performed transcriptome sequencing on TNBC cells with stable knockdown of PKMYT1 to clarify the specific mechanism *via* which PKMYT1 promoted TNBC cell proliferation. A total of 97 genes were significantly downregulated following PKMYT1 knockdown, while 59 genes were abnormally upregulated (Fig. 2A). Subsequent, GO analysis showed that these DEGs were significantly enriched in cholesterol metabolism (Fig. 2B). We observed significantly downregulated mRNA expressions of multiple enzymes involved in the cholesterol biosynthetic pathway and the expression of *SREBF2* (Fig. 2C). The results of the Western blot demonstrated that PKMYT1 knockdown greatly downregulated the protein expression of SREBF2, DHCR24, SQLE, FDFT1, and HMGCS.

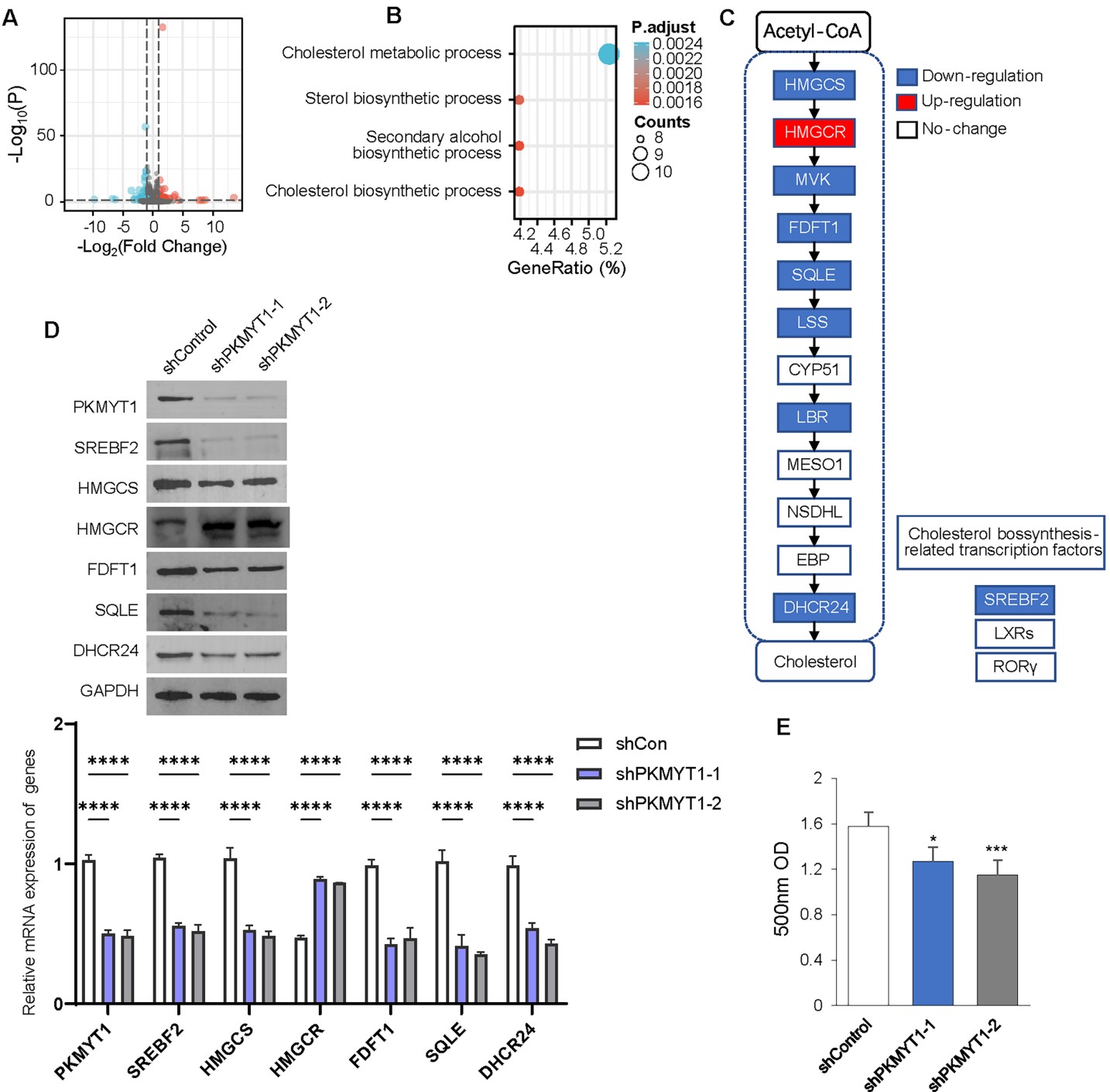

**Figure 2 PKMYT1 promotes cholesterol biosynthesis in TNBC.** (A) RNA extraction and sequencing were performed in MDA-MB-453-shPKMYT1-1 cells. Up-regulated (red) and downregulated (blue) mRNAs were characterized using volcano plots. (B) Differential genes were filtered under |Log2FC| > 0.5, FDR < 0.05 and subjected to GO pathway enrichment analysis. (C) mRNA expression of cholesterol biosynthesis-correlated enzymes and transcription factors, based on the above RNA-seq data of MDA-MB-453-shPKMYT1-1 cells. (blue presents down down-regulation; white presents no change). (D) Western blotting to demonstrate the changes in protein levels of cholesterol biosynthesis-related enzymes (HMGCS, HMGCR, FDFT1, SQLE, DHCR24) and transcription factor (SREBF2) after knockdown by PKMYT1 in MDA-MB-453 cells. (E) Total cholesterol content in MDA-MB-453-shPKMYT1-1/2 was determined by the Micro Total Cholestenone (TC) Content Assay Kit. $^*P < 0.05$, $^{***}P < 0.001$, $^{****}P < 0.0001$, ANOVA Test.

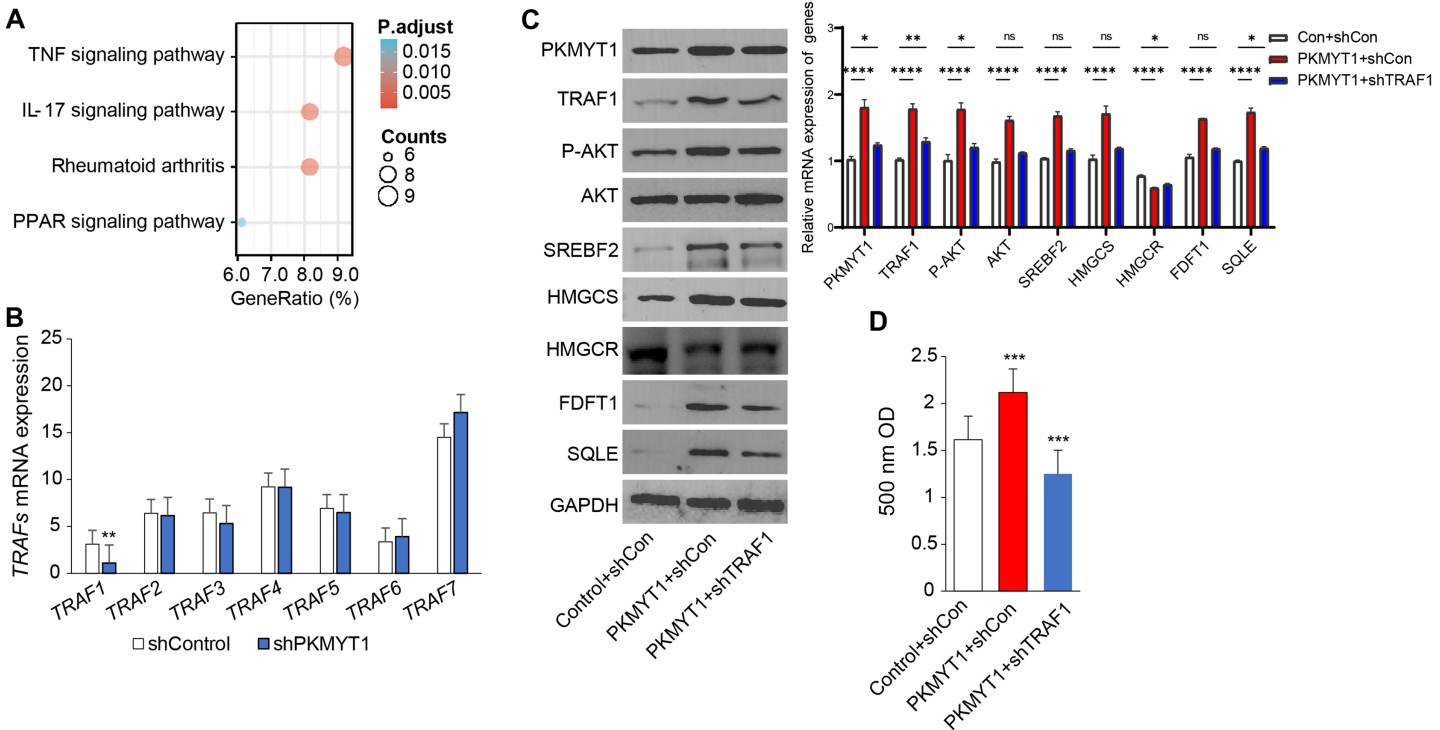

**Figure 3 PKMYT1 promotes SREBF2-mediated cholesterol biosynthesis *via* the TNF/TRAF1/AKT pathway.** (A) KEGG pathway enrichment analysis was performed on differential genes (|Log2FC| > 0.5, FDR < 0.05). (B) mRNA expression of TRAFs, based on the above RNA-seq data of MDA-MB-453-shPKMYT1-1 cells. $^{**}P < 0.01$, unpaired two-tailed Student's t test. (C) Western blotting of PKMYT1, TRAF1, P-AKT(Ser473), AKT, SREBF2, HMGCS, HMGCR, FDFT1, SQLE and GAPDH in MDA-MB-453 cells transduced with PKMYT1 or in combination with TRAF1 shRNA. Total protein was extracted after 100 ng/ml TNF treatment for 24 h. (D) The Micro Total Cholestenone (TC) Content Assay Kit was conducted to measure the total cholesterol content in MDA-MB-453 cells transduced with PKMYT1 or in combination with TRAF1 shRNA. $^{*}P < 0.05$, $^{***}P < 0.001$, $^{****}P < 0.0001$, ANOVA Test.

Notably, the protein expression level of HMGCR was upregulated after the knockdown of PKMYT1 (Fig. 2D). PKMYT1 knockdown significantly reduced the content of cholesterol in MDA-MB-453 cells (Fig. 2E). To conclude, our findings supported that PKMYT1 played a functional role in the regulation of cholesterol biosynthesis in TNBC cells.

## PKMYT1 promoted SREBF2-mediated cholesterol biosynthesis *via* the TNF/TRAF1/AKT pathway

To clarify the specific mechanism through which PKMYT1 promoted SREBF2-mediated cholesterol biosynthesis, we performed KEGG enrichment analysis based on the transcriptome data from PKMYT1 stable knockdown cell lines. It was found that the TNF signaling pathway was closely related to PKMYT1 knockdown (Fig. 3A). This relationship suggested that inhibition of PKMYT1 had potential effects on the TNF signaling pathway, which in turn may exert downstream effects on SREBF2-mediated cholesterol biosynthesis. Our study found that silencing PKMYT1 significantly downregulate *TRAF1* mRNA expression and overexpression of PKMYT1 significantly upregulated TRAF1 protein expression (Figs. 3B and 3C). Additionally, under TNF treatment conditions, PKMYT1 overexpression enhanced AKT activation and the expression of proteins related

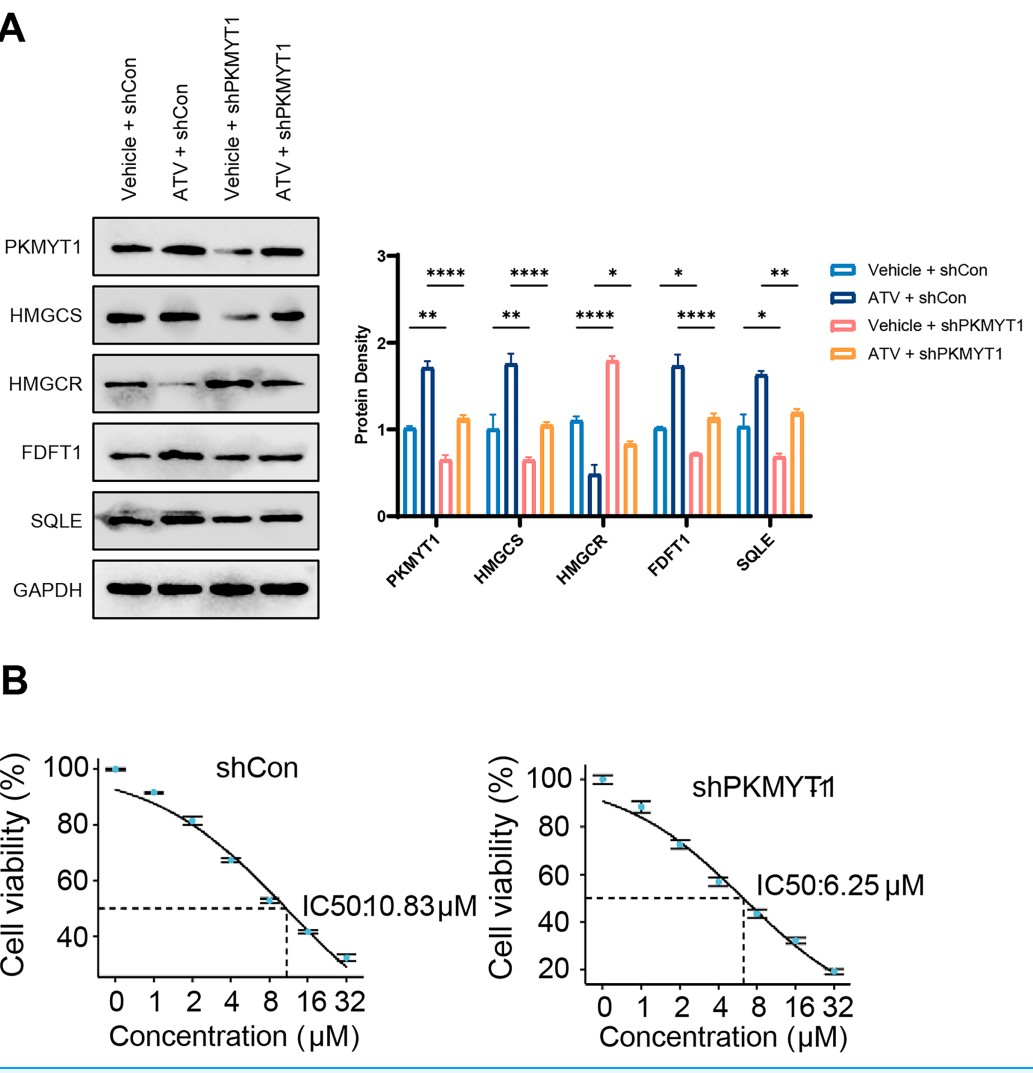

**Figure 4 PKMYT1 knockdown negates atorvastatin-induced feedback of cholesterol biosynthesis and up-regulates the drug sensitivity of atorvastatin.** (A) Western blotting of PKMYT1, HMGCS, HMGCR, FDFT1, SQLE and GAPDH in MDA-MB-453-shCon and sh-PKMYT1-1 cells treated with vehicle or atorvastatin (5 μM) for 24 h. (B) IC50 values of atorvastatin in MDA-MB-453-shCon and sh-PKMYT1-1 cells. $^*P < 0.05$, $^{**}P < 0.01$, $^{****}P < 0.0001$.

to cholesterol biosynthesis such as SREBF2, SQLE, HMGCS, FDFT1 and intracellular cholesterol content (Figs. 3C and 3D). However, knockdown of TRAF1 significantly inhibited the activation of AKT and cholesterol biosynthesis mediated by PKMYT1 overexpression (Figs. 3C and 3D). These findings indicated that the TNF/TRAF1/AKT/SREBF2 signaling pathway was essential for PKMYT1-induced cholesterol biosynthesis.

## PKMYT1 knockdown reversed atorvastatin-induced feedback of cholesterol biosynthesis and promoted the drug sensitivity of atorvastatin

The cholesterol biosynthesis pathway is an abundantly upregulated feedback pathway following statin treatment of TNBC cells (*Cai et al., 2019*; *Kimbung et al., 2016*; *Sarkar & Chattopadhyay, 2023*). As PKMYT1 knockdown inhibited SREBF2 expression and its

mediated cholesterol biosynthesis, we hypothesized that PKMYT1 knockdown can suppress statin-induced feedback activation of SREBF2. Notably, atorvastatin (ATV) treatment upregulated the protein expression of SQLE, FDFT1, HMGCS, and this feedback upregulation was significantly inhibited by PKMYT1 downregulation (Fig. 4A). In addition, as shown in Fig. 4B, in the control group, the response of cancer cell lines to different concentrations of ATV showed an IC50 value of 10.83 μM, whereas the IC50 of ATV was significantly lower than that of the control group when PKMYT1 was silenced (IC50 = 6.25 μM). This indicated that TNBC cells were more sensitive to ATV after PKMYT1 knockdown. Overall, these results confirmed that PKMYT1 knockdown eliminated the feedback upregulation of cholesterol biosynthesis-related proteins caused by statins and promoted the drug sensitivity of ATV.

## DISCUSSION

TNBC exhibits a greater proliferation index and a lack of effective therapeutic targets, in comparison to other breast cancer subtypes. Cholesterol, a crucial element in the structure of cell membranes, plays a vital role in controlling intracellular traffic and signaling pathways. Though cells typically absorb cholesterol from the blood, they have the ability to produce cholesterol themselves when there is a greater demand caused by tissue remodeling or abnormal cell growth, such as in cases of cancer (Coradini, 2024). It has been demonstrated that cholesterol biosynthesis is significantly enhanced in TNBC (Xu et al., 2020). Based on public databases and bioinformatics tools, some researchers have shown that genetic variants in cholesterol biosynthesis pathway genes are associated with prostate cancer risk (Cheng et al., 2021). Changes in cholesterol biosynthetic processes in TNBC may provide intriguing pharmacological targets and novel therapeutic strategies for TNBC. In this study, we identified PKMYT1, a member of the serine/threonine protein kinase family, as a critical driver of the cholesterol biosynthesis in TNBC cells. PKMYT1 was aberrantly high-expressed in TNBC and was related to an unfavorable prognosis. Furthermore, PKMYT1 activated the TNF/TRAF1/AKT signaling pathway, thereby upregulating the expression of enzymes involved in SREBF2-mediated cholesterol synthesis. Notably, knockdown of PKMYT1 significantly suppressed the feedback upregulation of the statin-mediated cholesterol synthesis pathway, and PKMYT1 knockdown increased the drug sensitivity of ATV in TNBC cells.

Cancer cells rely on cholesterol to meet their increasing nutritional requirements and to sustain their uncontrolled proliferation to promote tumor progression. Several clinical trials have shown that elevated circulating cholesterol levels are related to a quick breast cancer recurrence (Lettiero et al., 2018; Villa et al., 2016). Higher cholesterol biosynthesis is also one of the metabolic features of TNBC (Xu et al., 2020). In addition, researchers found that abnormally promoted cholesterol biosynthesis functions played a crucial role in the proliferation, survival and differentiation of breast cancer stem cells (Ehmsen et al., 2019). However, the mechanisms underlying the abnormal upregulation of cholesterol biosynthesis in TNBC remained unclear. SREBF2, LXRs, and ROR are transcription factors that regulate the cholesterol biosynthesis pathway (Cai et al., 2019). Our study revealed that PKMYT1 noticeably upregulated the mRNA and protein expression of

SREBF2. Studies have confirmed that multiple oncogenic pathways including PI3K-AKT, mTOR and c-Myc are capable of promoting SREBF2 transcriptional production (*Wang et al., 2017*; *Li et al., 2014*; *Chen et al., 2022*). In a previous study, PKMYT1 has been found to activate the AKT/mTOR pathway in esophageal squamous cancer (*Zhang et al., 2019*). Our study further identified the TNF-TRAF1-AKT pathway as a crucial pathway for PKMYT1 to regulate SREBF2 expression. Therefore, combined with the abnormally high expression of PKMYT1 in TNBC, PKMYT1 might be a key factor in promoting cholesterol biosynthesis in TNBC.

HMGCR is a rate-limiting enzyme in cholesterol biosynthesis, and statins are the most common pharmacological inhibitors of HMGCR. Targeting HMGCR is considered to be an effective clinical treatment strategy for cancer (*Tan et al., 2019*). Statins have been found to be able to prevent proliferation or induce apoptosis in a range of cancer cell lines (*e.g.*, breast, ovarian, and prostate tumor cells) *in vitro* (*Borgquist et al., 2018*; *Clendening & Penn, 2012*). However, clinical trials using statins in adjuvant therapy have largely failed to reduce the incidence risk of cancer. For instance, *Jouve et al. (2019)* demonstrated that incorporating pravastatin into a treatment plan with sorafenib cannot improve the survival of patients with advanced hepatocellular carcinoma. In a comparable study, *Kim et al. (2014)* observed that the addition of simvastatin to a capecitabine–cisplatin regimen did not improve progression-free survival for patients with gastric cancer. This may be related to the promotion of cholesterol biosynthesis in tumors by statins through feedback activation of SREBF2. Our study confirmed that PKMYT1 knockdown inhibited the statin-induced feedback activation of SREBF2. Significantly, knockdown of PKMYT1 increased the drug sensitivity of ATV in TNBC cells. Overall, our study revealed a novel function of PKMYT1 in TNBC, providing a novel target for targeting tumor metabolic reprogramming in TNBC. However, there were still some limitations in this research study. Our study used an *in vitro* cell culture model for mechanistic studies, but this may not fully reflect the complexity of the *in vivo* environment. Therefore, further validation of our results using animal models is needed. In addition, we explored the preliminary function of PKMYT1 in TNBC, and the detailed and precise molecular mechanism of its regulation on the SREBF2-mediated cholesterol biosynthesis pathway requires further investigation.

## ABBREVIATIONS

| | |
|---|---|
| **TNBC** | Triple-negative breast cancer |
| **PR** | Progesterone receptor |
| **ER** | Estrogen receptor |
| **SREs** | Sterol response elements |
| **RIPA** | Radioimmunoprecipitation assay |
| **SDS-PAGE** | Sodium dodecyl sulfate–polyacrylamide gel electrophoresis |
| **TC** | Total Cholestenone |

### Funding

The authors received no funding for this work.

### Competing Interests

The authors declare that they have no competing interests.

### Author Contributions

- Wei Gao conceived and designed the experiments, analyzed the data, authored or reviewed drafts of the article, and approved the final draft.
- Xin Guo conceived and designed the experiments, prepared figures and/or tables, and approved the final draft.
- Linlin Sun conceived and designed the experiments, analyzed the data, prepared figures and/or tables, and approved the final draft.
- Jinwei Gai performed the experiments, prepared figures and/or tables, and approved the final draft.
- Yinan Cao performed the experiments, analyzed the data, authored or reviewed drafts of the article, and approved the final draft.
- Shuqun Zhang performed the experiments, analyzed the data, authored or reviewed drafts of the article, and approved the final draft.

### Data Availability

The raw data is available in GitHub and Zenodo:

- https://GitHub.com/1shuqun/New-Raw-data-in-Revising.git

- 1shuqun. (2024). 1shuqun/New-Raw-data-in-Revising: My raw data in revising (v.1.1.1). Zenodo https://doi.org/10.5281/zenodo.11464343.

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
