# Peer review of "PKMYT1 knockdown inhibits cholesterol biosynthesis and promotes the drug sensitivity of triple-negative breast cancer cells to atorvastatin"

_PeerJ, doi:10.7717/peerj.17749_

## Round 0.1 · original submission · Major Revisions

While the study is interesting and potentially significant, there are several critical issues that need to be addressed before the manuscript can be considered for publication.

Reporting and Language:
(1) Ensure that abbreviations are defined at their first mention in the main text, even if they were defined in the abstract.
(2) Provide the manufacturer or supplier details for specific commercial products used in the study.
(3) Improve the overall English language to ensure clarity for an international audience.
Experimental Design and Methodology:
(1) Provide detailed experimental procedures for assays such as the EDU assay, Plate colony formation assay, and cholesterol measurement, to ensure transparency and reproducibility.
(2) Clearly describe the criteria and details of data screening and processing from the TCGA database.
(3) Clarify whether the statistical analysis of bioinformatics was also performed using SPSS 18.0 software.
(4) Explain the formula used for calculating clone forming efficiency in Figure 1F.
(5) Include scale bars in microscopic images to quantify the analysis.
(6) Discuss the limitations of the current study and provide suggestions for future research.
(7) Expand the introduction and background sections to better contextualize the study within the broader field and appropriately reference relevant prior literature.

Results and Discussion:
(1) Quantify all Western blots and provide representative images of control and knockdown cells to demonstrate the impact of AKT pathway activation on cellular morphology.
(2) Clarify the rationale for using drug sensitivity to explain the pathways and relationships of PKMYT1, and the use of IC50 to elucidate the pathways.
(3) Provide a more detailed explanation and literature review to support the discussion regarding cholesterol biosynthetic processes as potential pharmacological targets.
(4) Elucidate why PKMYT1 specifically promotes SREBF2-mediated cholesterol synthesis.
(5) Provide further evidence or a detailed explanation to support the claimed close relationship between TNF and PKMYT1 knockdown, as well as the connection between MPS1, Pi3K, and PKMYT1 in this context.
(6) Address the discrepancy between the mRNA expression level of TRAF1 in Figure 3 and the Western blot results.

**Language Note:** The Academic Editor has identified that the English language must be improved. PeerJ can provide language editing services - please contact us at [email protected] for pricing (be sure to provide your manuscript number and title). Alternatively, you should make your own arrangements to improve the language quality and provide details in your response letter. – PeerJ Staff

Reviewer 1 ·

Basic reporting

1. When an abbreviation is used for the first time in the main text, even if it has been defined in the abstract, it should be re-provided with the full name and the abbreviation in parentheses to ensure clarity and self-sufficiency. You mentioned "Triple-negative breast cancer (TNBC)" in your abstract, and you should have said "Triple-negative breast cancer (TNBC)" when you first mentioned it in the introduction.
2. In the Materials and Methods section of scientific papers, references to specific commercial products often require the manufacturer or supplier who provided the product for the sake of transparency and experimental reproducibility. However, no manufacturer or supplier was given for pBoBi expression vectors, Flag-tagged expression vectors and CCK-8.
3. The English language should be improved to ensure that aninternational audience can clearly understand your text.

Experimental design

4. The EDU assay, Plate colony formation assay, and Measure of cholesterol in the Materials and Methods section do not give a general experimental procedure, which is very important for the transparency and reproducibility of the study.
5. Ensuring comprehensive and clear methodological details is crucial for the scientific community's ability to evaluate, replicate, and extend research findings. The authors analyzed the data from the TCGA database, but the criteria and details of data screening and processing were not presented in the methods.
6. The authors conducted the statistical analysis of bioinformatics and the statistical analysis of cell experiments. Is the statistical analysis of bioinformatics also completed by SPSS 18.0 software? Please explain in the statistical analysis section.
7 In Figure 1F, the authors quantify cell proliferation by clone forming effiency, what is the formula for the calculation of this measure?
8. Scale is used to indicate how much a certain length in the picture represents the actual length. It is an important tool to express the size of the image and quantify the analysis in scientific research. A scale bar is indicated when showing pictures of experimental results of edu-stained cells.
9. Discussing the limitations section is an important part of honestly presenting the findings of the study and providing a comprehensive perspective. This not only helps the reader to correctly understand the scope and applicability of the study, but also shows the author's deep understanding and critical thinking of his own work. It is recommended to discuss the limitations of the current study and make suggestions for future research.

Validity of the findings

no comment

Additional comments

no comment

Reviewer 2 ·

Basic reporting

The article should include sufficient introduction and background to demonstrate how the work fits into the broader field of knowledge. Relevant prior literature should be appropriately referenced. Ensure all external claims are properly cited. Figures should be relevant to the content of the article, of sufficient resolution, and appropriately described and labeled.

Experimental design

Lines 199-203: Quantification of all Western blots is essential. The impact of AKT pathway activation on cellular morphology should be demonstrated with representative images of control and KD cells.
Line 207: The rationale for using drug sensitivity to explain the pathways and relations of PKMYT1 needs clarification.
Line 215: An explanation is needed for using IC50 to elucidate the pathways.
Lines 222-223: The two sentences are not linked; more explanation is required.
Lines 224-226: A literature review is recommended to contextualize the discussion regarding cholesterol biosynthetic processes as potential pharmacological targets.

Validity of the findings

Why does PKMYT1 specifically promote SREBF2-mediated cholesterol synthesis? More elucidation is required.
Lines 195-196: Further evidence or a detailed explanation is needed to support the claim of the close relationship between TNF and PKMYT1 knockdown.
Line 196: An explanation is required to establish the connection between MPS1, Pi3K, and PKMYT1 in this context.
Lines 197-198: The mRNA expression level of TRAF1 in Figure 3 does not align with the Western blot results; clarification is necessary.

Additional comments

- Provide specific details about the ANOVA test used.
- Clarify whether error bars represent SD or SEM.
- Ensure all external claims are properly cited.
In summary, while the study holds promise, the manuscript requires substantial revisions to address the outlined concerns and ensure its suitability for publication in its current state.

Annotated reviews are not available for download in order to protect the identity of reviewers who chose to remain anonymous.

Reviewer 3 ·

Basic reporting

1. Line 29: you use comparative words like ‘greater’ and ‘worse’, are you referring in comparison to other breast cancer subtypes? Please rephrase the sentence to reflect this.
2. Lines 47-49: Witten in a very confusing way. Triple-negative breast cancer is ER, PR negative with no Her2 amplifications. There is no TNBC with or without Her2 amplification.
3. Lines 49-51: The current statistics state that TNBC is 10-15% of all breast cancer cases. Please cite more recent papers reflecting this up-to-date statistic.
4. Line 60: The first use of an abbreviation should always be listed in full form first to introduce the abbreviation – in this case ‘MPS’.
5. Line 62: typo – change to ‘mutations’.
6. Lines 65-66, line 89, lines 105-106, line 129, line 139: Grammatically incorrect, please rephrase.
7. Line 79-80: Please cite the clinical trials or data showing that statins were ineffective as adjuvant therapy for cancer patients.
8. Line 82: Sudden transition - The authors provide no introduction at all to the WEE kinase family or any background of its significance in cholesterol synthesis or cancer. They also do not specify the abbreviation full form before first use.
9. Lines 84-86: Missing citation
10. Lines 252-254: Missing citation
11. Line 213: missing commas
12. Lines 209-210: Please provide more citations as this is claimed by the authors as a widely accepted fact so it should have more studies supporting it.
13. Lines 257-258: Missing citations. Please cite more than one clinical trial.

Experimental design

1. Line 100: The catalog number provided for the TRAF1 antibody is incorrect. Please fix this.
2. Line 107: Provide company source of FBS
3. Line 110: This section needs major work. Authors do not provide plasmid vector names and details as well as source. Transfection protocol details are also missing.
4. Line 113: Missing lot of details again. Why is this plasmid not mentioned in the plasmids section? The title of this section says stable cell line generation but text contains no information on which cell lines were transduced in this methodology. What was the viral transduction protocol used?
5. Line 120: Change title to ‘EdU Cell Proliferation assay’
6. Lines 124-126: Was the counting manual?
7. Lines 135-149: Missing lot of details - Was the library prep stranded or unstranded? What was the depth of sequencing? Was it paired-end or single end? And was it SE/PE 75 or 150? The authors also do not provide any details on the pipeline used for differential gene expression analysis.
8. Lines 154-157: Unclear. What is CCK-8? Instead of ‘various concentrations’ specify the concentration range/gradient used to assess this. Was a kit used at the endpoint – provide details.

Validity of the findings

1. Line 166: Do you mean ‘over-expressed’?
2. Figure 1A – unclear how the classifications were done. Structural variant refers to a collection of all other types of genetic alterations mentioned. What does this signify? Also, when you say ‘mutation’ in green, are you referring to single nucleotide point mutations or insertions?
3. Figure 1E – unclear how the data has been plotted. How is the colony formation efficiency determined to be ~12% for the control group?
4. Figure 2A – why was the RNA sequencing done on just one set of shRNA? It would be important to show both shRNAs to rule out any off-target impacts.
5. Figure 3B – it is unclear what is the data plotted. The Y-axis says mRNA expression, is it via qPCR or direct RNAseq data? If RNASeq, is it FPKM or TPKM?
6. Figure 4 – there are a lot of issues with this figure such as lack of proper controls and supporting information. This is one the most critical figures for the paper and hence it is very concerning that this is not convincing. Figure 4A should have more controls such as just ATV treatment without a shcontrol as well as a vehicle+siPKMYT1 control. This latter control will provide insights into whether PKMYT1 expression is increased by ATV treatment since the knockdown is not as strong as figure 2. All of the western results should also be quantified. The increase in HMGCR agrees with the reference #21 that the author’s cite for lines 209-210, however, it also disagrees with lines 252-254 in the discussion.

---

## Round 0.2 · accepted · Accept

All comments have been addressed by reviewers. This paper has been improved after revisions, and I think it can be accepted for publication.

Reviewer 1 ·

Basic reporting

The manuscript overall has good language expression, the citation section provides sufficient background information, the literature citations are appropriate, the quality of the figures and tables is high, and the descriptions in the captions are clear.

Experimental design

The manuscript provides sufficient foundational data, which is statistically reliable. The statistical power calculations of the experimental design show that the design is reasonable.

Validity of the findings

no comment

Additional comments

no comment

Reviewer 2 ·

Basic reporting

As long as the revised article has all the information according reviewer requested, and all the answers given by corresponding author also in the article cohesively, the article should be accepted.

Experimental design

Experimental design as of now demonstrate the claim, however, it is hardly explain the whole idea of metabolic reprogramming.

Validity of the findings

Additional images of cell morphology would strengthen the finding.